# Physical health in young males and risk of chronic musculoskeletal, cardiovascular, and respiratory diseases by middle age: A population-based cohort study

**Aleksandra Turkiewicz**[1]*, **Karin Magnusson**[1,2], **Simon Timpka**[3,4], **Ali Kiadaliri**[1], **Andrea Dell'Isola**[1], **Martin Englund**[1]

**1** Clinical Epidemiology Unit, Orthopedics, Clinical Sciences Lund, Lund University, Lund, Sweden, **2** Norwegian Institute of Public Health, Oslo, Norway, **3** Perinatal and Cardiovascular Epidemiology, Clinical Sciences Malmö, Lund University, Lund, Sweden, **4** Departments of Obstetrics and Gynecology Skåne University Hospital, Lund and Malmö, Sweden

* aleksandra.turkiewicz@med.lu.se

**Data Availability Statement:** All data used in this work are available from national Swedish registered after obtaining suitable ethical

## Abstract

### Background

Cardiovascular, respiratory, and musculoskeletal disease are among the leading causes of disability in middle-aged and older people. Health and lifestyle factors in youth have known associations with cardiovascular or respiratory disease in adulthood, but largely unknown associations with musculoskeletal disease.

### Methods and findings

We included approximately 40,000 18-year-old Swedish males, who completed their conscription examination in 1969 to 1970, followed up until age of 60 years. Exposures of interest were physical health: body mass and height, blood pressure, pulse at rest, muscle strength, cardiorespiratory fitness, and hematocrit; self-reported lifestyle: smoking, alcohol, and drug use; self-reported health: overall, headache and gastrointestinal. We followed the participants through the Swedish National Patient Register for incidence of common musculoskeletal (osteoarthritis, back pain, shoulder lesions, joint pain, myalgia), cardiovascular (ischemic heart disease, atrial fibrillation), and respiratory diseases (asthma, chronic obstructive pulmonary disease, bronchitis). We analyzed the associations using general estimating equations Poisson regression with all exposures included in one model and adjusted for parental education and occupation. We found that higher body mass was associated with higher risk of musculoskeletal (risk ratio [RR] per 1 standard deviation [SD] 1.12 [95% confidence interval, CI 1.09, 1.16]), cardiovascular (RR 1.22 [95% CI 1.17, 1.27] per 1 SD) and respiratory diseases (RR 1.14 [95% CI 1.05, 1.23] per 1 SD). Notably, higher muscle strength and cardiorespiratory fitness were associated with higher risk of musculoskeletal disease (RRs 1.08 [95% CI 1.05, 1.11] and 1.06 [95% CI 1.01, 1.12] per 1 SD difference in exposure), while higher cardiorespiratory fitness was protective against both

permission and complying with data protection laws in Sweden. The data from conscription examination are available from Riksarkivet (https://riksarkivet.se/), the data from the Population Register and socioeconomic variables are available from Statistics Sweden (https://www.scb.se/vara-tjanster/bestall-data-och-statistik/mikrodata/), the data from the National Patient Register and the Causes of Death Register are available from The National Board of Health and Welfare (https://bestalladata.socialstyrelsen.se/data-for-forskning/). The code for the statistical model is included in supplementary file.

**Funding:** The study was supported by grants from Greta and Johan Kock Foundation (AT) (http://www.kockskastiftelsen.se/stiftelserna.html), the Swedish Research Council (ME and AD) (https://www.vr.se), the Swedish Rheumatism Association (ME) (https://reumatiker.se/), and Governmental Funding of Clinical Research within National Health Service (ALF) (ME) (https://www.intramed.lu.se/forska/alf-anslag-och-ansokan), Österlund Foundation (ME) (https://www.alfredosterlundsstiftelse.se/), and the Foundation for People with Movement Disability in Skåne (ME) (https://www.stiftbistandskane.se/). The funders had no role in the data collection, study design, conduct, reporting or decision to publish.

**Competing interests:** ME declares consultancy for Grünenthal Sweden AB and Key2Compliance. AK acts as a part-time scientific advisor for Joint Academy providing a digital self-management program for musculoskeletal pain. AT is associate editor for statistics at Osteoarthritis and Cartilage journal. ST was supported by a career grant from the Swedish Research Council (2019-02082). Other authors declare no conflict of interest.

**Abbreviations:** CI, confidence interval; ICD, International Classification of Diseases; ISCO, International Standard Classification of Occupations; RR, risk ratio; SD, standard deviation.

cardiovascular and respiratory diseases (RRs 0.91 [95% CI 0.85, 0.98] and 0.85 [95% CI 0.73, 0.97] per 1 SD exposure, respectively). We confirmed the adverse effects of smoking, with risk ratios when comparing 11+ cigarettes per day to non-smoking of 1.14 (95% CI 1.06, 1.22) for musculoskeletal, 1.58 (95% CI 1.44, 1.74) for cardiovascular, and 1.93 (95% CI 1.60, 2.32) for respiratory diseases. Self-reported headache (category "often" compared to "never") was associated with musculoskeletal diseases (RR 1.38 [95% CI 1.21, 1.58]) and cardiovascular diseases (RR 1.29 [95% CI 1.07, 1.56]), but had an inconclusive association with respiratory diseases (RR 1.13 [95% CI 0.79, 1.60]). No large consistent associations were found for other exposures. The most notable associations with specific musculoskeletal conditions were for cardiorespiratory fitness and osteoarthritis (RR 1.23 [95% CI 1.15, 1.32] per 1 SD) and for muscle strength and back pain (RR 1.18 [95% CI 1.12, 1.24] per 1 SD) or shoulder diseases (RR 1.27 [95% CI 1.19, 1.36] per 1 SD). The main limitations include lack of adjustment for genetic factors and environmental exposures from childhood, and that the register data were available for males only.

## Conclusions

While high body mass was a risk factor for all 3 studied groups of diseases, high cardiorespiratory fitness and high muscle strength in youth were associated with increased risk of musculoskeletal disease in middle age. We speculate that these associations are mediated by chronic overload or acute trauma.

## Author summary

### Why was this study done?

- Chronic diseases of the heart, lung, and musculoskeletal system are the leading causes of disability in middle-aged and older people.

- Health and lifestyle factors in youth are known to be associated with heart and lung disease later in life, but little is known of the relationship with common musculoskeletal conditions such as osteoarthritis, shoulder-, and back pain diagnoses.

### What did the researchers do and find?

- We studied about 40,000 Swedish male conscripts who did their conscription examination in 1969 to 1970 at 18 years of age.

- Information from the conscription examination included body mass and height, blood pressure, pulse at rest, muscle strength, physical fitness, smoking, alcohol and drug use.

- Using national Swedish healthcare register data of diagnostic codes set by physicians in specialist and inpatient care, the conscripts were followed until the age of 60 years for common conditions of the heart, lung, and musculoskeletal system.

- We found that high body mass as well as tobacco smoking in youth were linked with increased risk for all the studied diseases later in life.

- Notably, a high level of physical fitness and high muscle strength in youth were associated with increased risk of musculoskeletal disease, primarily osteoarthritis, and back pain.

### What do these findings mean?

- Our findings reinforce that maintaining normal body mass and non-smoking are beneficial in the prevention of common chronic diseases including conditions of the musculoskeletal system.

- The associations of physical fitness with musculoskeletal diseases require validation and further studies of mediating mechanism, i.e., the events between youth and middle age that could lead to disease.

- We speculate that the associations with musculoskeletal diseases later in life are driven by chronic overload and (or) joint injuries. Thus, better musculoskeletal injury prevention, e.g., in popular organized sports with high risk of joint injury, needs further attention.

- The main limitation is that based on the data source, we were only able to include males in the study.

## Introduction

Chronic musculoskeletal diseases are among the top reasons for years lived with disability in middle-aged and older persons, closely following cardiovascular and respiratory diseases [1]. Cardiovascular and respiratory diseases are the leading causes of mortality and disability and they are highly prevalent in middle-aged persons [2,3]. In contrast to musculoskeletal diseases, their early life factors have been studied extensively [4,5]. For example, healthy body mass and good respiratory fitness are known to promote cardio-respiratory health, while smoking and extensive alcohol consumption are well-established risk factors [4,6]. Thus, these diseases form a useful contrast outcome when studying musculoskeletal disease.

The most prevalent musculoskeletal diseases—which include back pain, osteoarthritis, shoulder lesions, myalgia, and unspecific joint pain—have pain as the main symptom and often uncertain or complex structural and molecular pathogenesis. The management of these musculoskeletal diseases is mainly reactive as treatment starts when symptoms are manifest, at which point it may often be too late to slow down, stop, or reverse the disease process [7]. Theory about disease etiology points to a "mismatch" between modern environmental conditions and our genetic makeup. This discrepancy is often linked to reduced physical activity and altered dietary habits as compared with our ancestors, leading to low grade inflammation [8]. Biomechanical reasons could interplay in the development of these diseases [9,10]. Assessing risk factors in youth is especially important as musculoskeletal diseases develop slowly, and thus many potential risk factors measured in middle-age can instead be early consequences of the already ongoing disease process.

Sweden, a country with comprehensive nationwide health care registers, offers an opportunity to study health outcomes over long follow-up times [11]. Combined with detailed

exposure data from conscription, which was mandatory for all males aged 18 years and included standardized objective measures of physical health, these registers offer a unique data source for studying physical health in youth as a risk factor for health outcomes in middle age [12]. Previously, such studies have typically focused on studying associations between specific exposures and outcomes. For example, cannabis use and schizophrenia [13], blood pressure and cardiovascular events [14], overweight or smoking and mortality [15], muscle strength and knee osteoarthritis or joint pain [16,17].

The study of physical health in youth and risk of later disease could potentially reveal a "window of opportunity" for preventative strategies to delay or hinder disease onset. We aimed to take a comprehensive look at the health factors in youth and the occurrence of chronic diseases in middle age. The exposures of interest include objectively measured physical health indicators: body size, cardiorespiratory fitness, blood pressure and pulse at rest, hematocrit, and muscle strength; as well as self-reported lifestyle and health: smoking, use of alcohol and drugs, gastrointestinal problems, headache, and overall health. We aimed to estimate their associations with the incidence of musculoskeletal, cardiovascular, and respiratory diseases 30 years later. We further aimed to estimate their associations with the most common types of musculoskeletal diseases, i.e., back pain, osteoarthritis, shoulder lesions, myalgia, and joint pain.

## Methods

### Study cohort

We obtained data on Swedish male conscripts born in the second half of 1950 or in 1951 who underwent conscription in the years 1969 and 1970, typically at age 18 years. During this time, the data collection included measurements of physical fitness and a lifestyle questionnaire for collecting data on self-reported smoking, alcohol consumption, use of drugs and overall health, all collected prior to the start of the military service. We identified 40,307 males who had conscription examination and answered the questionnaire. The total number of conscripts in these years is not exactly known, but it is estimated that around ~5,000 persons did the examination but did not answer the lifestyle questionnaire and their data are not available to us (personal communications with State Archives [Riksarkivet]). Further, we included persons alive and resident in Sweden at January 1, 1987, which is the start of the current study, $n = 40,294$. We linked the identified conscripts' data with: the Population Register for demographic information, the National Patient Register for information about registered diagnoses in specialist outpatient and inpatient care, the Causes of Death Register, and retrieved information on parental education and occupation. We used encrypted personal identification number for the linkage.

### Exposures

From the medical examination at conscription, we retrieved variables related to physical health and lifestyle as detailed in Table 1. The measurements were done according to standardized protocols [18].

Cardiorespiratory fitness (physical work capacity) was tested by means of the maximal work test, where the tested person is required to work, until reaching the point of exhaustion, on a bicycle ergometer with an invariable, heavy load. The load was chosen individually with respect to constitution and prior history of physical activity. Thus, we use the term "cardiorespiratory fitness" to denote this variable (maximal power output, unit Watt) [20]. One could argue that many of the continuous exposures measured objectively could describe similar underlying physical composition and thus could be combined. We have performed principal

**Table 1. Variables related to physical health assessed at conscription examination and used as exposures in the current study.**

| Exposure | Description |
|---|---|
| Physical health—measurements | |
| Body mass | Kilograms (kg) |
| Body height | Centimeters (cm) |
| Blood pressure | Average of systolic and diastolic blood pressure in mmHg |
| Pulse at rest | Number of beats per minute |
| Hematocrit | Volume percentage of red blood cells in blood |
| Muscle strength | Standard measure calculated as 1.7*hand grip strength+1.3*knee extension strength+0.8*elbow flexion strength, reported in Newtons. |
| Cardiorespiratory fitness | Provided in Watt categorized into 9 groups. In analyses, we treated this variable as continuous, with values set to the middle of the range in each category. |
| Lifestyle factors—self-reported | |
| Use of drugs | Yes or no (reference category) |
| Smoking | Categorical, Non-smoker (reference category), 1–5, 6–10, or 11+ cigarettes per day |
| Alcohol use | Categorized into none (reference category), 1–100 gram per week, 101+ grams per week, based on self-reported amount of beer, wine, and spirit used, according to previously developed formulas [19]. |
| Health-related factors—self-reported | All included as categorical variables |
| Overall health | 5-item Likert scale, from "very good" to "very bad" |
| Headache | 4-item Likert scale from "yes, often" to "no, never" |
| Gastrointestinal problems | 4-item Likert scale from "yes, often" to "no, never" |

component analysis including these 7 variables. We found that we would need 6 components to retain at least 90% of variability in the 7 exposures and thus we kept the 7 variables as separate exposures of interest. The correlations of the 7 variables are provided in Table A in S1 Supporting information.

## Outcomes

We studied 3 disease groups: musculoskeletal, respiratory, and cardiovascular disease based on diagnoses registered in the Swedish National Patient Register according to the Swedish translations of ICD-9 and ICD-10 from 1987 to 2010 (Table 2). Within cardiovascular and respiratory disease, we selected diagnoses that were most prevalent and registered mainly as underlying diagnosis at the healthcare visit. Within the musculoskeletal diseases, we selected non-inflammatory chronic pain conditions that have poorly understood etiology and no disease

**Table 2. International Classification of Diseases (ICD)-9 and ICD-10 codes for selected diseases.**

| Disease group | Disease | ICD-9 | ICD-10 |
|---|---|---|---|
| Musculoskeletal | Shoulder lesions | 726* | M75 |
| | Myalgia | 729 | M79 |
| | Osteoarthritis of peripheral joints | 715 | M15-M19 |
| | Joint pain | 719E | M25.5 |
| | Back pain | 723, 724 | M54 |
| Respiratory | Asthma, COPD†, Bronchitis | 493, 496, 491 | J45-J46, J44, J40-J42 |
| Cardiovascular | Ischemic heart disease, Atrial fibrillation | 410–414, 427D | I20-I25, I48 |

*ICD-9 code of 726 could include other joints than shoulder.

†COPD–chronic obstructive pulmonary disease.

modifying treatment, have high prevalence in the modern society and in this cohort, were registered mainly as the underlying cause at the healthcare visit, and are known leading causes of life years lived with disability. Thus, we included back pain, osteoarthritis, shoulder lesions, myalgia, and joint pain. The register includes data on physical healthcare visits in specialist out-patient and all in-patient care for all residents of Sweden [11,21]. We retrieved available data with national coverage, i.e., from visits in in-patient care from years 1987 to 2010, one-day surgery 1997 to 2010 and out-patient specialist care 2001 to 2010. We considered a person to have a disease (binary outcome) if having the relevant diagnostic code registered at least once between 1987 and 2010, as the register did not have national coverage prior to the year 1987. A person who died during the follow-up time and had one of our diseases of interest registered as a cause of death (using the same ICD-9 or 10 codes) was considered to have the disease. Persons with incomplete follow-up (due to death from other causes or emigration) were assumed to not have the diseases of interest if having none of the relevant diagnostic codes registered prior to loss to follow-up.

To avoid reversed causality, we excluded males who had at least one of the diseases of interest already at the conscription examination. Up to 6 diagnoses per individual were registered at the conscription examination according to the ICD-8 system (Table B in S1 Supporting information).

## Statistical analysis

We provided descriptive statistics for all exposures and outcomes, as means with standard deviations or frequencies and percentages. For primary analysis, we included the 3 disease groups as outcomes, i.e., we treated all included musculoskeletal diseases as one outcome.

We used the Poisson regression model with generalized estimating equations (GEE) with log link for analysis and estimation of risk ratios (Stata's command *xtpoisson*). The model can be fitted with all outcomes in one model, while accounting for the correlation of the outcomes within persons. We assumed unstructured correlation in the GEE model. The data set was structured as for fitting models for multiple unordered outcomes of different types, i.e., each person had 1 data row for each of 3 outcomes and outcome type (musculoskeletal, cardiovascular, or respiratory disease) was included as a categorical variable.

The cohort was homogenous with respect to age and sex. We did not assume any specific causal structure between the exposures, as they were all measured at the same time and could affect each other in either direction. Thus, the modeling strategy was to fit univariable models (including mass and height always together, as they describe body size), a model adjusted for body size (mass and height), and a model including all exposures. The last model was additionally adjusted for: (a) father's occupation (classified into professionals, administrative or clerical workers, sales workers or unknown or no occupation, farmers and fisherman, miners and workers in transport, craftsmen, production or service or military) based on occupation registered according to Swedish version of International Standard Classification of Occupations (ISCO)-58 classification; (b) mother's occupation (classified into having occupation or not, as 78% of mothers did not work outside of home); (c) highest education level attained by mother (binary, at least secondary school or lower). We did not additionally adjust for paternal education, as it was highly correlated with occupation. We adjusted for parental characteristics as we consider them to be true confounders, i.e., preceding both the exposures and outcomes. In each model, we included the outcome type to allow for different prevalences of each outcome and an interaction between each exposure and outcome type to allow for potentially different associations between each exposure and each outcome. No interaction of outcome type with confounders was included, as the socioeconomic factors have the same direction of associations with all 3 studied disease groups [22]. In a secondary analysis, we analyzed each

musculoskeletal condition as a separate outcome using a Poisson regression model, adjusted as above. We avoided using body mass index in the main model, due to its complicated statistical (a ratio variable) and epidemiological interpretations [23,24], but we used it in a sensitivity analysis and to enable comparisons with other studies.

The estimates for continuous variables are reported per 1 standard deviation (SD) difference in exposure to make the estimates comparable. Cardiorespiratory fitness was available to us in categorized form and thus we could not calculate SD. Thus, we report the estimates per 50 Watt which corresponds to SD reported in another study utilizing data from conscription examination in Sweden [25]. We checked that the assumption of linearity was reasonable for the continuous exposures, by categorizing them into deciles and plotting estimates versus mean values of exposures in each created category. The linearity holds within the range of observed exposure values and should not be extrapolated outside of this range.

Given that we adjusted for all physical exposures at the same time, the estimates need to be interpreted thereafter, i.e., an estimate represents the change in risk associated with one unit difference in a specific exposure, all other things being equal. We presented all estimates (risk ratios [RRs]) with 95% confidence intervals (CIs). We did not correct for multiple testing, as we avoid categorizing the results into significant or not and we consider this observational study to be hypothesis generating [26].

Prompted by comments from reviewers we included several sensitivity analyses. We included results adjusted for BMI (instead of body mass and height), and adjusted for pulse at rest or cardiorespiratory fitness, but not both in the same model. Further, as cardiorespiratory fitness is often expressed normalized by body mass (W/kg) we fitted models with such exposure (Methods - normalizing cardiorespiratory fitness by body mass in S1 Supporting information). We did not use this variable in our main analyses due to known interpretational problems when using ratio variable, which is a statistical interaction [23,27]. We fitted both models without and with main effects of both variables.

We included figures of marginal means by exposure levels for the 3 outcomes for illustrative purposes. These were derived from the fitted Poisson GEE models using Stata's *margins* command (CIs calculated using delta method). All analyses were done in Stata 18, 2023. Stata Statistical Software: Release 18. College Station, TX: StataCorp LLC.

### Handling of missing data

Of the persons included in the analysis, 19% (6,847) had at least 1 missing value in the baseline variables. Of these, 53% had only 1 missing value and an additional 31% had only 2 values missing. We speculate that the data is missing completely at random, as the registration at conscription examination was routine and there are no clear patterns among the missing data. However, to check if excluding persons with missing data could affect our results, we used multiple imputation (Methods –Multiple imputation in S1 Supporting information).

The study was approved by Ethical Review Board in Lund (Dnr 2011/500). Given that the project reuses existing register data, no active consent was required while an option to "opt-out" was available.

This study is reported as per the Reporting of Studies Conducted using Observational Routinely Collected Data (RECORD) guideline (S1 Checklist).

## Results

### Descriptive data

We identified 40,294 males that were alive on January 1, 1987. We excluded 4.8% who had musculoskeletal disease (most often back pain), 1.9% who had respiratory disease, and 0.4%

who had cardiovascular diseases already at the conscription examination. We followed the remaining 37,476 persons until December 31, 2010, i.e., age of 59 to 60 years.

At the conscription examination, 83% of males reported having very good or quite good overall health, while 59% were smokers and 6% declared not using alcohol. Mean mass and height were 66 kg and 178 cm (Table 3). Only 0.8% had a body mass index above 30.

The cumulative incidence of musculoskeletal diseases diagnosed between 1987 and 2010 was 20%, cardiovascular 10%, and respiratory diseases 3%. Among 12,821 persons with at least one of these 3 diseases, 83% had only one of the included diseases. Among persons without any of the diseases, 1,585 died and 594 emigrated from Sweden during the study time.

## Associations with the musculoskeletal, respiratory, and cardiovascular diseases

The unadjusted associations between the exposures and the different outcomes (disease groups) are presented in Table C in S1 Supporting information, the adjusted associations in Table 4 and results from sensitivity analyses in Tables D, E, and F in S1 Supporting information. Differences in adjustment variables made little difference to most of the estimates for physical health measurements (Fig A in S1 Supporting information) and the results from sensitivity analyses were similar to the main findings (Tables D, E, and F in S1 Supporting information). Higher muscle strength (by 1 SD) was associated with higher risk of musculoskeletal disease, RR 1.08 (95% CI 1.05, 1.11). This corresponds to approximately 1.5% absolute difference, with cumulative incidence of 21.7% in those with 1 SD higher muscle strength than average, and 18.7% in those with 1 SD lower muscle strength (Fig 1). Cardiorespiratory fitness was associated with higher risk of musculoskeletal disease (RR 1.06 [95% CI 1.01, 1.12]) but lower risk of cardiovascular and respiratory diseases (Fig 1 and Table 4). The coefficient for cardiorespiratory fitness normalized by body mass versus musculoskeletal disease, measuring the relationship of both variables with the disease risk, was 0.96 (95% CI 0.93, 0.98) (Tables I and J in S1 Supporting information). Higher body mass, irrespective of all other included measures of physical health, was associated with 12% to 22% increased risk of all 3 outcomes. In absolute numbers, cumulative incidence of musculoskeletal disease increases by approximately 2.5% with 1 SD change in body mass, resulting in marginal incidence proportion of 22.6% in persons with 1 SD higher mass than average compared to 17.9% in those with 1 SD lower mass than average (Fig 1). Consistently, the association between height and the outcomes had the opposite direction.

Higher blood pressure was associated with higher risk of cardiovascular disease with no or weak associations with the other diseases (Table 4). Lower pulse at rest was associated with lower risk of musculoskeletal and cardiovascular diseases but not respiratory diseases. We found no evidence for associations between hematocrit and the three studied outcomes. Results from multiply imputed data were essentially the same (Table H in S1 Supporting information).

When considering the lifestyle factors impacting health, smoking was associated with approximately 2 times higher risk of respiratory disease and 58% higher risk of cardiovascular disease in persons smoking 11+ cigarettes per day compared to non-smokers. Neither drug use nor alcohol intake had clear associations with the outcomes. Interestingly, we found no strong associations between self-reported health or gastrointestinal problems and future diagnosis of chronic diseases, after controlling for all the other variables (Table 4). Confidence intervals for the associations between headache and the 3 diseases were wide precluding strong conclusions. However, the risk ratios comparing the category "yes, often" to "no, never" were above one for associations with both musculoskeletal and cardiorespiratory diseases.

**Table 3. Descriptive data of exposures of interest at the conscription examination.**

| Exposure | Mean (SD) or *n* (%) |
|---|---|
| Body mass, kg | 66.5 (9.2) |
| Body height, cm | 178 (6) |
| Body mass index†, kg/m$^2$ | 21.0 (2.6) |
| Blood pressure, mmHg (mean of systolic and diastolic) | 100 (9) |
| Pulse at rest, beats per minute | 75.2 (12.9) |
| Hematocrit, % | 46.6 (2.5) |
| Muscle strength, *N* | 2,021 (297) |
| Cardiorespiratory fitness (Watt, Watt per kg*), *n* (%) | |
| Up to 189, 3.07 | 2,246 (6) |
| 190–209, 3.30 | 5,279 (14) |
| 210–224, 3.40 | 8,778 (23) |
| 225–239, 3.51 | 7,048 (19) |
| 240–254, 3.65 | 4,398 (12) |
| 255–269, 3.82 | 3,636 (10) |
| 270 and above, 3.91 | 6,089 (16) |
| Used drugs, *n* (%) | 3,936 (11) |
| Smoking (cigarettes per day), *n* (%) | |
| Non-smoker | 15,189 (41) |
| 1–5 | 4,190 (11) |
| 6–10 | 7,863 (21) |
| 11 or more | 9,679 (26) |
| Alcohol, grams per week, *n* (%) | |
| None | 2,200 (6) |
| 1–100 | 24,306 (67) |
| 101 or more | 9,575 (27) |
| Overall health, *n* (%) | |
| Very good | 15,882 (43) |
| Quite good | 15,362 (41) |
| Neither good nor bad or worse | 5,927 (16) |
| Headache, *n* (%) | |
| Often | 1,598 (4) |
| Sometimes | 8,916 (24) |
| Occasionally | 20,217 (54) |
| Never | 6,420 (17) |
| Gastrointestinal problems, *n* (%) | |
| Often | 1,526 (4) |
| Sometimes | 5,355 (14) |
| Occasionally | 17,695 (48) |
| Never | 12,532 (34) |

*Mean value in each group of cardiorespiratory fitness.

†Body mass index was included in the sensitivity analyses.

SD, standard deviation.

## Associations with specific musculoskeletal diseases

Higher body mass was associated with all included specific musculoskeletal diseases (Fig 2 and Table G in S1 Supporting information). The weakest (or inconclusive) associations were with

**Table 4. Risk ratios (95% CIs) for associations between exposures and musculoskeletal, cardiovascular, and respiratory diseases.** All exposures were included in 1 model. Adjusted for parental occupation and education.

| Exposure | Musculoskeletal | | Cardiovascular | | Respiratory | |
|---|---|---|---|---|---|---|
| Mass, per 1 SD | 1.12 | [1.09, 1.16] | 1.22 | [1.17, 1.27] | 1.14 | [1.05, 1.23] |
| Height, per 1 SD | 0.96 | [0.94, 0.99] | 0.92 | [0.88, 0.95] | 0.91 | [0.85, 0.98] |
| Blood pressure, per 1 SD | 0.98 | [0.95, 1.00] | 1.11 | [1.07, 1.16] | 0.98 | [0.91, 1.06] |
| Pulse at rest, per 1 SD | 0.93 | [0.90, 0.96] | 0.95 | [0.91, 0.99] | 0.98 | [0.91, 1.06] |
| Hematocrit, per 1 SD | 1.01 | [0.98, 1.03] | 1.02 | [0.98, 1.06] | 1.06 | [0.99, 1.13] |
| Muscle factor, per 1 SD | 1.08 | [1.05, 1.11] | 1.01 | [0.97, 1.06] | 0.98 | [0.90, 1.06] |
| Cardiorespiratory fitness, per 50 Watt | 1.06 | [1.01, 1.12] | 0.91 | [0.85, 0.98] | 0.85 | [0.73, 0.97] |
| Used drugs | 1.03 | [0.95, 1.12] | 1.04 | [0.92, 1.16] | 1.09 | [0.89, 1.34] |
| Did not use drugs (reference) | 1.00 | | 1.00 | | 1.00 | |
| Non smoker (reference) | 1.00 | | 1.00 | | 1.00 | |
| 1–5 cigarettes | 0.96 | [0.88, 1.05] | 1.13 | [0.99, 1.28] | 1.27 | [0.99, 1.63] |
| 6–10 cigarettes | 1.04 | [0.97, 1.12] | 1.30 | [1.18, 1.44] | 1.41 | [1.16, 1.72] |
| 11+ cigarettes | 1.14 | [1.06, 1.22] | 1.58 | [1.44, 1.74] | 1.93 | [1.60, 2.32] |
| Alcohol: none | 0.97 | [0.87, 1.09] | 1.02 | [0.87, 1.20] | 1.26 | [0.94, 1.70] |
| Alcohol: 1–100 gram per week (reference) | 1.00 | | 1.00 | | 1.00 | |
| Alcohol: 101 or more grams per week | 1.09 | [1.03, 1.16] | 0.93 | [0.85, 1.01] | 1.02 | [0.87, 1.20] |
| Health: Very good (reference) | 1.00 | | 1.00 | | 1.00 | |
| Health: Quite good | 0.96 | [0.90, 1.01] | 0.96 | [0.88, 1.04] | 0.91 | [0.78, 1.07] |
| Health: Neither good nor bad | 0.94 | [0.86, 1.02] | 1.01 | [0.90, 1.13] | 1.17 | [0.95, 1.43] |
| Headache: yes, often | 1.38 | [1.21, 1.58] | 1.29 | [1.07, 1.56] | 1.13 | [0.79, 1.60] |
| Headache: yes, sometimes | 1.18 | [1.09, 1.29] | 1.07 | [0.94, 1.21] | 1.06 | [0.83, 1.34] |
| Headache: yes, occasionally | 1.10 | [1.02, 1.18] | 1.11 | [1.00, 1.23] | 1.04 | [0.85, 1.27] |
| Headache: no, never (reference) | 1.00 | | 1.00 | | 1.00 | |
| Stomach problems: yes, often | 1.12 | [0.99, 1.28] | 1.09 | [0.91, 1.31] | 1.21 | [0.88, 1.68] |
| Stomach problems: yes, sometimes | 1.05 | [0.96, 1.14] | 1.04 | [0.93, 1.17] | 1.12 | [0.90, 1.40] |
| Stomach problems: yes, occasionally | 1.01 | [0.95, 1.07] | 0.99 | [0.91, 1.08] | 1.00 | [0.85, 1.18] |
| Stomach problems: no, never (reference) | 1.00 | | 1.00 | | 1.00 | |

SD, standard deviation.

joint pain and myalgia. For myalgia, all estimates apart from those for body mass and hematocrit were close to 1. All confidence intervals for estimates of associations with joint pain were wide. Higher cardiorespiratory fitness, after adjusting for other exposures, seemed to be associated with higher risk of osteoarthritis (RR 1.23 [95% CI 1.15, 1.32] per 1 SD), while not as clearly with other included musculoskeletal diseases. Higher pulse at rest was associated with lower risk of osteoarthritis (RR 0.91 [95% CI 0.87, 0.94] per 1 SD), but not with back pain. In contrast, the risk ratios of associations between higher muscle strength and shoulder lesions or back pain were 1.27 (95% CI 1.19, 1.36) and 1.18 (95% CI 1.12, 1.24), respectively, while the estimates were smaller in magnitude for osteoarthritis, myalgia, or joint pain (Fig 2 and Table G in S1 Supporting information). Alcohol use and smoking, as well as self-reported health measures had generally similar associations with all included musculoskeletal diseases (Table G in S1 Supporting information). For example, smoking 11+ cigarettes per day compared to non-smoker had risk ratios of 1.20 (95% CI 1.07, 1.34) with myalgia, 1.03 (95% CI 0.94, 1.12) with osteoarthritis, 1.19 (95% CI 1.02, 1.39) with shoulder disease, 1.30 (95% CI 1.15, 1.47) with back pain, and 1.12 (95% CI 0.93, 1.35) with joint pain.

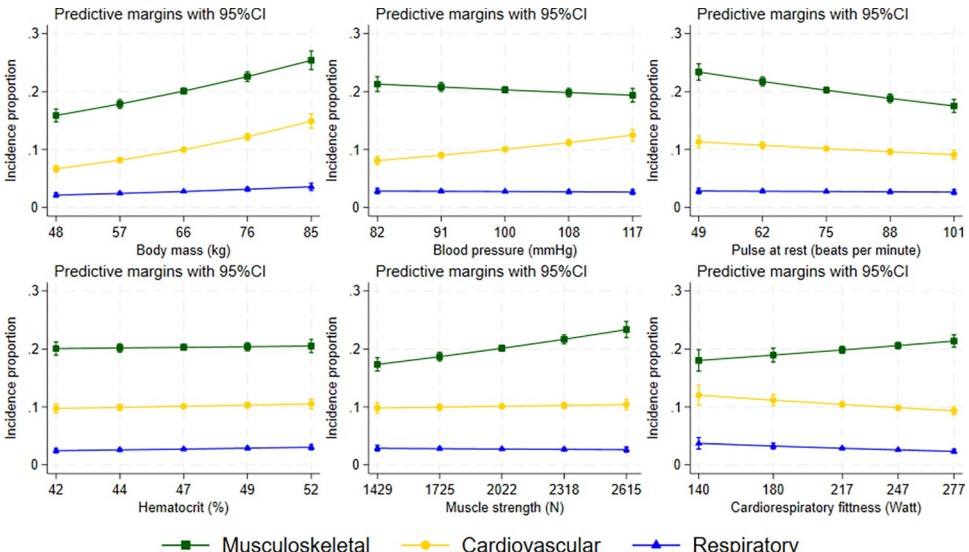

**Fig 1. Fitted incidence proportion by measures of physical health.** Marginal estimates from the adjusted Poisson regression model. All exposures (the 6 variables in the figure, height, drug use, smoking, alcohol consumption, overall health, headache, and gastrointestinal problems) were included in the model and they were also adjusted for parental education and occupation. The middle value on the x-axis corresponds to the mean value (all exposures apart from cardiorespiratory fitness) in the study sample and exposures are plotted over a range of 2 standard deviations on either side of this. For cardiorespiratory fitness, the values correspond to mid points of the intervals (where the middle value on the x-axis is the mode). Incidence proportion refers to the proportion of persons with diseases diagnosed during the follow-up period from 1987 to 2010. Musculoskeletal: osteoarthritis, back pain, shoulder lesions, joint pain, myalgia; Cardiovascular: ischaemic heart disease, atrial fibrillation; Respiratory: asthma, chronic obstructive pulmonary disease, bronchitis. CI, confidence interval.

## Discussion

We conducted a longitudinal study of approximately 40,000 males followed from exposures assessed at age 18 years to chronic musculoskeletal, respiratory, and cardiovascular diseases

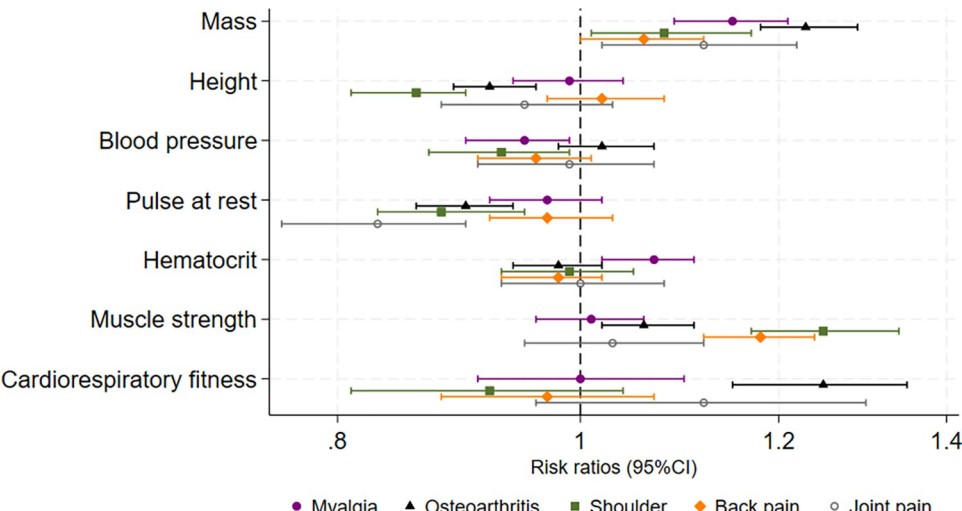

**Fig 2. Risk ratios (95% CIs) for the associations between 7 exposures related to physical health and specific musculoskeletal diseases.** All exposures included in 1 model, additionally adjusted for parental education and occupation. Estimates are per 1 standard deviation of exposure.

diagnosed up to age of 60 years. We report high cumulative incidence of diagnosed musculo-skeletal disease already by middle age, with one in 5 of the included persons affected. While high body mass is a risk factor for all 3 studied diseases, high cardiorespiratory fitness (physical work capacity) and high muscle strength in youth are associated with increased risk of muscu-loskeletal disease in middle age but decreased risk of cardiovascular or respiratory diseases. The results for musculoskeletal conditions are to our knowledge novel, as early risk factors of musculoskeletal conditions are understudied. Further, our study provides estimates of associations of physical health at youth with common chronic diseases from the same population and data sources, enabling a direct comparison.

High muscle strength in youth seems not protective against future musculoskeletal disease but is rather a risk factor. Similar results were reported for associations between muscle strength in adolescence and self-reported musculoskeletal pain by middle age [17], and we reported similar associations between knee extensor strength and knee osteoarthritis specifi-cally [16]. This is in contrast with most estimates reported from studies assessing muscle strength in closer proximity to the incidence of these conditions or in cross-sectional designs [28,29]. We speculate that low muscle strength in middle age could already be a consequence of the slowly developing disease and not a risk factor.

We found that higher cardiorespiratory fitness in adolescence is associated with higher risk of musculoskeletal disease by middle age. There are at least 2 potential explanations for these findings. One is unmeasured confounding from unknown factors causing both high physical fitness and musculoskeletal problems, for example genetics. The other one is related to the mediating effect of acute joint injury and repetitive overload. It is possible that extensive physi-cal activity, either due to sports participation or choice of physically demanding occupations, may lead to injuries or overload that manifest as or lead to musculoskeletal pain in middle age. Another important factor to consider is that in the main analyses, we used cardiorespiratory fitness measure from physical work capacity test, not normalized by body mass. Such normal-ized cardiorespiratory fitness is often a good predictor of performance, but may be less suitable for describing epidemiological associations, as it represents both relationships with cardiore-spiratory fitness and inverse of body mass. When the 2 relations are in opposite directions (as in our data), such a combined estimate may be difficult to interpret [20,23].

Higher body mass is an independent risk factor for all included outcomes, irrespective of muscle strength, cardiorespiratory fitness, and other included exposures such as smoking and alcohol consumption. Our data reinforce the notion that maintaining low body mass is impor-tant for promoting cardiovascular, respiratory, and musculoskeletal health in middle age [30–33]. Importantly, our estimates are not driven by severe obesity, as only <1% of included con-scripts had body mass index over 30, and are in line with earlier reports of a linear increase in risk of osteoarthritis with increasing BMI even in the normal range (BMI between 18 and 25) [34]. Other exposures with consistent associations with all 3 chronic diseases are smoking and self-reported headache. While the adverse effects of smoking are well known, associations between headache and migraine with other chronic conditions lack a known biological path-way [35]. Increased pain sensitivity as a consequence of headache has been proposed [36].

In our secondary analysis of specific musculoskeletal diagnoses, most of the estimated asso-ciations for physical exposures were similar in direction and magnitude considering the uncer-tainty reflected by the confidence intervals. However, notable differences occur. We cannot say if these differences in estimates are due to sampling variation (and reporting many esti-mates) or represent true differences in disease etiology. The associations with the less specific diagnosis of myalgia are the weakest. Estimates for joint pain are generally consistent with those for osteoarthritis, but more uncertain. Considering that joint pain is a common initial

diagnosis when a specific cause of the pain has not yet been confirmed, these results seem logical.

Higher cardiorespiratory fitness and lower pulse at rest seem to be associated specifically with diagnosis of osteoarthritis. This finding could be consistent with increased risk of post-traumatic disease due to participating in sports requiring good cardiorespiratory fitness, such as football or handball [37].Unfortunately, there exist no register data covering the period after conscription examination but before middle age, when most of joint injuries typically occur [38]. Observational evidence that joint injury is a strong risk factor for future osteoarthritis is strong, while genetic contribution to both cannot be ruled out [39–41]. High muscle strength has the strongest associations with back and shoulder lesions. Physical stress has been reported consistently as risk factor for back pain [42], which could explain the association. We did not find any other clear associations with back pain, but a previous study also based on Swedish conscript data found support for association between low cardiorespiratory fitness and disability pension due to musculoskeletal problems including back pain [43]. There is a paucity of longitudinal studies of risk factors for shoulder lesions, with uncertain evidence of perceived muscle tension in transition from school to working life [44].

There are important limitations that should be considered when interpreting our results. First, estimates come from adjusted regression models, where all exposures were included simultaneously. Thus, the estimates represent the association assuming "all other things being equal"—depending on the true causal structure, these could be closer to the total or direct effect. We were not able to disentangle the causal structure between the exposures, given that they were all measured at the same time and are interrelated. However, together they provide a relevant picture of a young person's health state. We were not able to adjust the regression models for early life factors, which could be potential confounders [45]. The most important include birth weight and type and intensity of physical activity. Low birth weight is associated with lower physical fitness and higher incidence of chronic diseases [46]. We cannot exclude confounding from genetic factors, as moderate heritability has been reported for musculoskeletal disease, and is more uncertain for physical fitness [47–49]. We did not have primary care data, which means that patients managed exclusively in primary care would be misclassified. Consequently, we focused on conditions with many patients consulting in specialist and inpatient care, and those that are typically the main reason for the visit (i.e., registered as main diagnosis). Therefore, hypertension was not considered within cardiovascular diseases, as it would not be reliably captured in the register data. Another potential source of bias is censoring due to death or emigration, but in this relatively young cohort this was uncommon (approximately 5% of the study sample), and the causes of death were typically distinct from our diseases of interest and included cancer, suicide, and accidents. Importantly, only <1% of included males were obese (with body mass index above 30). This is not representative of today's population of young males, and thus our results may not generalize to males suffering from obesity in adolescence. Our results may not be generalizable to females.

Early risk factors for musculoskeletal diseases should be studied more extensively. Adverse effects of physical activity in adolescents are unfortunately understudied [50]. The 2020 WHO guidelines on physical activity in adolescents do not include any study with acute injuries as outcome, despite injuries being identified by the expert panel as an outcome of critical importance and their high prevalence in this age group [51–53]. Injury prevention programs and rehabilitation after an injury have become standard of care during the 2000s [54,55]. Evaluating the effectiveness of these programs on the prevention of musculoskeletal disease at middle age should be a priority.

We report associations between higher cardiorespiratory fitness or muscle strength and increased risk of musculoskeletal disease in middle age. Given the high prevalence and burden

of musculoskeletal disease, these findings need to be followed up by life-course mediation analyses quantifying the role of joint injuries and physical stress related to occupational or leisure time activities. Our findings reinforce that maintaining low body mass could be beneficial in the prevention of chronic disease in middle age.

## Supporting information

**S1 Checklist. The RECORD statement–checklist of items, extended from the STROBE statement that should be reported in observational studies using routinely collected health data.**
(DOCX)

**S1 Supporting information.** Table A. Correlations (Pearson's) between the 7 continuous exposures. Table B. International Classification of Diseases (ICD)-8 diagnostic codes for identification of diseases at conscription examination. Table C. Unadjusted risk ratios (95% confidence intervals) for associations between exposure and outcomes. Body mass and height were included in 1 model. Table D. Associations, risk ratios (95% confidence intervals), between exposures and outcomes adjusted for body mass and height. Table E. Associations, risk ratios (95% confidence intervals), between exposures and outcomes from fully adjusted model, when using body mass index as a measure of body size, instead of height and mass. Table F. Associations, risk ratios (95% confidence intervals), when removing pulse at rest, cardiorespiratory fitness or both these variables from the fully adjusted model. Table G. Associations (risk ratios with 95% confidence intervals) between exposures and specific diseases within MSK spectrum. Methods–Multiple imputation. Table H. Risk ratios (95% confidence intervals) for associations between exposures and the 3 chronic diseases from multiply imputed data. All exposures were included in 1 model. Additionally adjusted for parental occupation and education. Methods–normalizing cardiorespiratory fitness by body mass. Table I. The crude risk ratios (RR) with 95% confidence intervals (CI) for associations between cardiorespiratory fitness normalized by body mass (W/kg), with and without main effects of the interaction variables. Table J. The adjusted risk ratios (RR) with 95% confidence intervals (CI) for associations between cardiorespiratory fitness normalized by body mass (W/kg), with and without main effects of the interaction variables. Code. Stata code for the main analysis model. Fig A. Risk ratio (95% confidence intervals) estimates with different sets of adjustment. Each outcome is presented in a separate panel.
(DOCX)

## Acknowledgments

We thank Velocity Hughes for initial input on the study design and manuscript text.

## Author Contributions

**Conceptualization:** Aleksandra Turkiewicz, Karin Magnusson, Simon Timpka, Ali Kiadaliri, Andrea Dell'Isola, Martin Englund.

**Data curation:** Simon Timpka, Martin Englund.

**Formal analysis:** Aleksandra Turkiewicz.

**Funding acquisition:** Aleksandra Turkiewicz, Martin Englund.

**Methodology:** Aleksandra Turkiewicz, Karin Magnusson, Simon Timpka, Ali Kiadaliri, Andrea Dell'Isola.

**Resources:** Martin Englund.

**Supervision:** Martin Englund.

**Writing – original draft:** Aleksandra Turkiewicz.

**Writing – review & editing:** Aleksandra Turkiewicz, Karin Magnusson, Simon Timpka, Ali Kiadaliri, Andrea Dell'Isola, Martin Englund.

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
