## [Editor Report · Decision Letter 0]

5 Jul 2024

Dear Dr Turkiewicz, 

Thank you for submitting your manuscript entitled "Physical health in young males and risk of chronic musculoskeletal, cardiovascular and respiratory diseases in middle age – population-based cohort study" for consideration by PLOS Medicine.

Your manuscript has now been evaluated by the PLOS Medicine editorial staff and I am writing to let you know that we would like to send your submission out for external peer review.

Please re-submit your manuscript within two working days, i.e. by Jul 09 2024.

Feel free to email me at atosun@plos.org or us at plosmedicine@plos.org if you have any queries relating to your submission.

Kind regards,

Alexandra Tosun, PhD

Associate Editor

PLOS Medicine

---

## [Decision Letter · Decision Letter 1]

19 Sep 2024

Dear Dr Turkiewicz,

Many thanks for submitting your manuscript "Physical health in young males and risk of chronic musculoskeletal, cardiovascular and respiratory diseases in middle age – population-based cohort study" (PMEDICINE-D-24-02132R1) to PLOS Medicine. We would like to thank you for your patience. We recognise that the review process has taken longer than usual. The paper has been reviewed by subject experts and a statistician; their comments are included below and can also be accessed here: [LINK]

As you will see, the reviewers found the manuscript to be interesting, but they raised a number of methodological concerns. After discussing the paper with the editorial team and an academic editor with relevant expertise, I'm pleased to invite you to revise the paper in response to the reviewers' comments. We plan to send the revised paper to some or all of the original reviewers, and we cannot provide any guarantees at this stage regarding publication.

We ask that you submit your revision by Oct 10 2024. However, if this deadline is not feasible, please contact me by email, and we can discuss a suitable alternative.

Don't hesitate to contact me directly with any questions (atosun@plos.org). 

Best regards, 

Alexandra 

Alexandra Tosun, PhD 

Associate Editor

PLOS Medicine

atosun@plos.org

Comments from the editorial team:

The manuscript prompted a discussion about the data and the conditions listed under the musculoskeletal umbrella. We wondered if it would be possible to explore the data in more depth and with more specificity, particularly for musculoskeletal conditions. Please note that it will be important to address Reviewer #2's concerns about the analysis very carefully. We also suggest highlighting and discussing more clearly what the study adds to existing knowledge.

Comments from the reviewers: 

Reviewer #1: Statistical review

This paper uses a longitudinal study of Swedish men to explore associations between various measures of physical health at age 18 and development of cardiovascular, respiratory and musculoskeletal diseases later in life.

I had some, mostly minor, comments on the statistical methods and reporting (although point 3 is potentially an important one to consider).

1. Abstract: I would recommend, if possible, providing CIs for each outcome reported in the abstract. Since a lot of associations are not reported in the abstract, it might be useful to clarify 'there were no other significant associations' (if that is the case).

2. Page 3 - can it be clarified that the assessments were done at the beginning of conscription? If done at the end, then much less variability in fitness would presumably be expected.

3. Statistical analysis: if someone died during follow-up, was the follow-up time included in the model via an offset? I am unsure of what assumption is being made if someone died of one of the causes of interest (e.g. cardiovascular disease) in terms of competing risks: I presume it is equivalent to non-informative censoring if the follow-up time is incorporated into an offset.

4. Results - it might be useful to add the proportion of subjects who had the three different outcomes of interest at the follow-up.

5. Table 4 - I'd recommend using 'height' instead of 'length'

James Wason

Reviewer #2: This is an interesting analysis of the association between factors related to physical health at age ~18 years and risk of cardiovascular, respiratory and musculoskeletal disease outcomes up to age 60 years. Associations between body mass and cardiorespiratory fitness with cardiovascular and respiratory diseases were in the expected directions (body mass associated with higher risk and CRF with lower risk). However, slightly unexpectedly, CRF was assocated with higher risk of musculoskeletal disease. However examination of the data suggests that this effect may be an artifact of the statistical analysis model. In the model both CRF (assessed as power output on a cycle ergometry test) and resting heart rate were both included in the statistical model. Resting heart rate is a measure of fitness in its own right, so essentially the model has two different measures of CRF. The associations of CRF (as assessed by power) and of resting heart rate on musculoskeletal disease were essentially equal but in opposite directions (0.93 for resting heart rate, 1.06 for CRF assessed by power). This suggests that the true association between CRF and muscloskeletal disease is actually close to null. If this is the case, the whole premise and interpretation of the data will completely change. I suggest that the authors repeat their analysis excluding resting heart rate from the model to see if the adverse association of CRF assessed by power with musculoskeletal disease persists if they do this. They should also remove CRF assessed by power from the model, in a separate analysis to see whether the benefical association of resting heart rate with musculoskeletal disease persists. It is likely that the entire interpretation of the data in the discussion will need to be rewritten in light of these analyses.

Further more minor comments include:

1) It would be helpful to include BMI rather than height and weight in the model

2) It would be helpful to express CRF in W/kg, rather than in W as this relates more to VO2max measured in ml/kg/min

3) It would be helpful to see the relationship between muscular strength and the outcomes in a model which does not also include CRF or resting heart rate. 

4) In table 4 and in places in the text, replace 'length' with 'height' 

Reviewer #3: This is a well-conducted and well-written study which produces some interesting results, particularly on the contrasting associations of early-life fitness and strength on cardiovascular versus musculoskeletal conditions in middle age. I very much hope to see it published soon. There are a couple of statistical issues where I would like to see sensitivity analyses in the supplement to reassure me (and the reader of the published article) that some non-obvious analytical choices have not greatly influenced the findings. The first is the choice to analyse linearly-adjusted body mass rather than BMI (see major issue below) and the second is the presentation of mutually-adjusted results for all exposures (see minor issue Results, para 4 below). Other than that, I have listed in the minor issues every place I could see where I thought the clarity of the manuscript could be improved. These are numerous but mostly very trivial and easily remedied.

Major issues:

BMI is commonly used to measure adiposity, rather than adjusting analyses of body mass for the person's height as you have done. Linear adjustment for height may not be the best approach since we expect body mass to increase more than linearly with height from simple geometry. Adjusting for height rather than using a ratio with a standard exponent also makes the assumption (rightly or wrongly) that the true medically-relevant measure of mass-for-height is independent of height. If you believe you have a good justification for your approach, rather than using BMI, it should be given in the manuscript. Furthermore, unless you have a really convincing argument against using BMI at all, I would like to see a sensitivity analysis in the supplement showing (hopefully) that analysis of BMI produces much the same conclusions as your analysis of body mass adjusted linearly for height.

Minor issues:

Abstract: Add "people" to "middle-aged and elderly people".

Abstract: In the results, rephrase the reporting of the body mass results to the style used for other exposures, so that the direction of the association is made explicit.

Abstract: The units of the risk ratios (per SD) are missing when they are reported for body mass.

Intro, para 1: In the first sentence, the words "years lived with" are unnecessarily specific and I would delete them.

Intro, para 1: Should be "middle-aged persons", not "middle-age persons"

Intro, para 3: "which was mandatory", not "that was mandatory"

Intro, para 3: In "they offer a unique data source for studying…", "they" is ill-defined. Replace it with "these registers" if that's what you meant. Later in the sentence, there's also a missing "a" in "as a risk factor" .

Intro, para 3: I don't think "OA" has been defined or expanded yet. I don't think the abbreviation is used again, so I would just write it out in full.

Methods, para 1: "who underwent conscription", not "that underwent conscription". It's probably not necessary to specify "the year" in "the year 1950" (etc).

Methods, para 1: I would write "during this time", not "during this year" to describe examinations in 1969 and 1970. If it's only part of each year, adding up to a one year period, please specify this.

Methods, para 1: There's a missing "in" in "The total number of conscripts in these years…"

Methods, para 1: You report later that you also excluded people who had an outcome in 1987 or who died before this time - this would be best reported here, along with any other exclusions.

Table 1: Be a bit more informative in the description of Hematocrit - percentage of what?

Table 1: For Cardiorespiratory fitness - how was the power output measured?

Table 1: At the end of the alcohol use section there's a comma which should be a full stop.

Methods, para 7: State here that you used a log link.

Methods, para 7: The description of the data structure is a bit ambiguous. Perhaps writing "one data row" instead of "one row with data" could help. At present, it could be interpreted as saying that there was one row per person, containing data for all three outcomes (which I don't think was the case).

Methods, para 8: When you write "length" here, do you mean "height"? "Length" is also used in some tables and supplementary tables. I suggest using "height" throughout to avoid confusion.

Methods, para 8: When you write "Finally", it could be taken to mean that adding the interaction between exposures and outcome type was an additional adjustment set separate from the three already described. I think you must in fact have done it in every model, given the combination of data for all three outcomes and the need to report separate RR. Please can you clarify this? Also, were the adjustment variables in the third adjustment set (parents' occupation, etc) also allowed to interact with outcome type? This would give a closer analogy to the results you'd get from separate models for each outcome because the associations with these adjustment variable might well be different for different outcomes but if you don't interact them you're forcing, for example, father's occupation to have the same association with all three outcomes. Please clarify and justify your choice.

Methods, para 9: It's clear from the results tables that categorical variables were modelled as categories relative to a reference category, but it would be helpful to confirm that here (i.e. that you didn't dichotomise them or treat them as continuous)

Results, para 3: Please state that these incidences are for the period 1987-2010, rather than just until 2010.

Table 3: Please present the exposures in the same order in all Tables & Figures. This just makes it easier to read.

Table 3: I think those means with only two significant figures (i.e. body mass, pulse and hematocrit) and their corresponding SD should be reported to one decimal place for a little more precison.

Table 3: I think you report the mean and SD of hematocrit, in units of % (RBC) but "(%)" is written in brackets like the % for categorical variables. Other units for continuous variables are given after a comma, so shouldn't this be the case for hematocrit? Also, pulse presumably has units of minute^-1 (i.e. per minute) - this should be added to the table.

Results, para 4: I find it confusing and incomplete to report the 95% CI without the estimate and without saying that they are 95% CI. Rather than "5-11%", please write 8% (95% CI: 5%, 11%) instead (and similarly for the other results reported in this way). 

Results, para 4: You refer to the unadjusted / partially adjusted results in Online supplementary tables 2 and 3, but you only say that they also varied in those tables. I think the reader will want to know whether the alternative adjustments made a big or small difference to the estimates - this helps us to know whether they are likely to be confounding/mediating each other. Eyeballing the supplementary tables, it looks to me as if results are broadly similar whichever adjustment set you use, but it would be much easier to see if you could add plots, similar to Figure 2 but with the different adjustment sets in place of the different musculoskeletal conditions, to the supplement so that the reader can quickly compare the unadjusted, mass&height-adjusted and fully-adjusted estimates. You should also briefly report on how influential the choice of adjustment was, in the main manuscript Results section.

Results, para 4: The supplementary tables themselves are titled "supplementary table X" - unless you are following a particularly misguided journal style, it would be clearer if they could be referred to in the same way in the text.

Table 4: Because the Exposure descriptions wrap lines, it's sometimes hard to see which numbers correspond to which Exposures. Indenting wrapped lines might help, and/or including risk ratios of 1 (no CI) for the reference categories (as you do in the supplement).

Results, para 5: These risk ratios have been considerably rounded. Please either report them to the same precision they are in Table 4 ("1.93 times higher" and "58% higher") or precede them with "about" or "almost".

Figure 1: Why is height excluded, when it is included in all other results? Please either include it or justify the omission.

Figure 1: In the figure text, please add the word "fitted" at the beginning to make it clear that these are fitted values (rather than observed values making a remarkably good linear approximation). I think the phrase "incidence proportion" also needs a bit of explanation to help the figure stand alone - it's the proportion of people developing the condition during the period 1987-2010, isn't it?

Figure 1: Again in the figure text, it took me a while to understand the last sentence. I think it's the "increments" that are ambiguous. Perhaps refer to tick marks and/or points instead? Or you could even use different symbols - something like "hollow symbols indicate the mean of each exposure, with filled symbols at increments of one standard deviation". 

Figure 1: Colour-blind people may struggle to read this graph - could you perhaps use different line types or different symbol shapes in addition to the colours?

Figure 2: Please use consistent names for the exposures and present them in a consistent order. There's a typo in "fittness".

Figure 2: Although it's theoretically possible to guess the colour code from the ordering, giving the different musculoskeletal conditions different symbol shapes (on the graph and in the legend) might be helpful for people with colour blindness.

Discussion, para 3: It should be "…that manifest as or lead to…"

Discussion, para 3: The final sentence ("Evaluating the effectiveness…") is very long and difficult to follow. Please can you re-phrase it? It might be best broken into two sentences.

Discussion, para 4: Should be "…earlier reports of a linear increase…" and "…even in the normal range"

Discussion, para 4: Please give the values of BMI which define the normal range.

Discussion, para 5: "less specific" would be better than the double negative "more unspecific"

Discussion, para 7: The reference to adjustment in the limitations section emphasises that it would be useful to have a brief report in the main manuscript and easy-to-read report in the supplement, to show whether the different adjustment sets considered here affected estimates.

Discussion, para 7: Could you perhaps include a correlation matrix of the exposures in the supplement, to further inform consideration of the relationships between them ?

Discussion, para 7: State why you might want to adjust for early life factors (presumably, as potential confounders).

Discussion, para 7: Another limitation to mention here is that you excluded people with one outcome condition in 1987 from the analysis of the others. You report that most people with one or more conditions had only one, but there's still potential for this to have biased estimates somewhat towards the null. You could check for this by doing single outcome analyses with and without the people who were excluded for having other outcomes.

Discussion, para 7: In terms of generalisation, the male-only data must also be a limitation.

Supplementary Table 3: Should be "…adjusted for body mass and height".

---

* Please upload any figures associated with your paper as individual TIF or EPS files with 300dpi resolution at resubmission; please read our figure guidelines for more information on our requirements: http://journals.plos.org/plosmedicine/s/figures. While revising your submission, please upload your figure files to the PACE digital diagnostic tool, https://pacev2.apexcovantage.com/. PACE helps ensure that figures meet PLOS requirements. To use PACE, you must first register as a user. Then, login and navigate to the UPLOAD tab, where you will find detailed instructions on how to use the tool. If you encounter any issues or have any questions when using PACE, please email us at PLOSMedicine@plos.org.

* DATA AVAILABILITY: If the data are not freely available, please describe briefly the ethical, legal, or contractual restriction that prevents you from sharing it. Please also include an appropriate contact (web or email address) for inquiries (again, this cannot be a study author).

FIGURES AND TABLES

SUPPLEMENTARY MATERIAL

REFERENCES

* Where website addresses are cited, please include the complete URL and specify the date of access (e.g. [accessed: 12/06/2024]).

STUDY TYPE-SPECIFIC REQUESTS

* Abstract: Please include the study design, population and setting, number of participants, years during which the study took place (enrollment and follow up), length of follow up, and main outcome measures.

* Please ensure that the study is reported according to the RECORD guideline (available from https://www.record-statement.org) and include the completed checklist as Supporting Information. Please add the following statement, or similar, to the Methods: "This study is reported as per the Reporting of Studies Conducted using Observational Routinely-Collected Data (RECORD) guideline (S1 Checklist)." When completing the checklist, please use section and paragraph numbers, rather than page numbers.

* For all observational studies, in the manuscript text, please indicate: (1) the specific hypotheses you intended to test, (2) the analytical methods by which you planned to test them, (3) the analyses you actually performed, and (4) when reported analyses differ from those that were planned, transparent explanations for differences that affect the reliability of the study's results. If a reported analysis was performed based on an interesting but unanticipated pattern in the data, please be clear that the analysis was data driven. 

* Please state in the Methods section whether the study had a prospective protocol or analysis plan. If a prospective analysis plan (from your funding proposal, IRB or other ethics committee submission, study protocol, or other planning document written before analyzing the data) was used in designing the study, please include the relevant document(s) with your revised manuscript as a Supporting Information file to be published alongside your study and cite it in the Methods section. A legend for this file should be included at the end of your manuscript. If no such document exists, please make sure that the Methods section transparently describes when analyses were planned, and when/why any data-driven changes to analyses took place. Changes in the analysis, including those made in response to peer review comments, should be identified as such in the Methods section of the paper, with rationale.

---

## [Decision Letter · Decision Letter 2]

22 Oct 2024

Dear Dr Turkiewicz,

Many thanks for submitting your manuscript "Physical health in young males and risk of chronic musculoskeletal, cardiovascular and respiratory diseases in middle age – population-based cohort study" (PMEDICINE-D-24-02132R2) to PLOS Medicine. The paper has been reviewed by subject experts and a statistician; their comments are included below and can also be accessed here: [LINK]

Thank you for your detailed response to the editors' and reviewers' comments. As you will see, the reviewers are mostly satisfied with your responses to their comments. You will see that both subject-matter reviewers ask you to repeat/adjust certain analyses, which the editorial team agrees with and asks you to modify or add as additional analyses as suggested by the reviewers. Please also note that reviewer #3 feels that some of their points, although addressed in the rebuttal, were not actually implemented in the manuscript. After discussing the paper with the editorial team and an academic editor with relevant expertise, we ask you to carefully address the comments in a further revision. We plan to send the revised paper to some or all of the original reviewers.

We ask that you submit your revision by Nov 12 2024. However, if this deadline is not feasible, please contact me by email, and we can discuss a suitable alternative.

Don't hesitate to contact me directly with any questions (atosun@plos.org). 

Best regards, 

Alexandra 

Alexandra Tosun, PhD 

Associate Editor

PLOS Medicine

atosun@plos.org

Comments from the reviewers: 

Reviewer #1: Thank you to the authors for addressing my previous comments (and those of the other reviewers). I have no further issues to raise.

Reviewer #2: Thank you for making these changes which have improved the manuscript. I appreciate the effort in this comprehensive revision, which has addressed most of my concerns. I would still like you to repeat the analysis expressing CRF in terms of W/kg (both with and without resting heart rate in the model). VO2max or maxiumum power expressed per kg body mass is the classic way of expressing CRF as this is the metric which is most strongly associated with indices of physical function/performance (e.g. 10K running time). A 50kg person with maximum power of 150W and a 100kg person with maximum power of 300W would have similar running performance. So W/kg is the appropriate unit for CRF here (and the effects on performance are the same whether this goes up via increasing W or decreasing kg). The fact that CRF expressed in W is not the most appropriate metric is clear from the relatively strong correlation with body mass (r = 0.39). This is counterintuitive for a measure of CRF - the fastest people in endurance events tend to be the lightest, not the heaviest people. Expressing CRF in W/kg may also change the strength of association with resting heart rate. Could you please undertake these analyses in a revised version?

Reviewer #3: While many of my comments have been addressed, the revisions appear to have been done rather carelessly, contrasting with the high quality of the previous version of the manuscript. In several places, you claim in your response to have applied (generally minor) changes suggested by me, but on reading the manuscript, you have not. This is very annoying to a reviewer.

Nonetheless, my major concern has been mostly addressed and I like the study. If you can address my remaining comments, I think it should be published. I would like to see BMI analysed in SD in the sensitivity analysis, like the other exposures are, and there is an apparent inconsistency in the follow-up period which needs to be resolved. My other comments are matters of language and clarity to the reader.

Minor issues

Abstract: Colon, not full stop after "Methods and Findings"

Abstract: The abstract still doesn't say which way round the associations with body mass were, doesn't distinguish the three outcomes and doesn't give the CI of the RR. These should be reported the same way as other results in the abstract, e.g. "Higher body mass was associated with higher risk of musculoskeletal (risk ratio X.XX per standard deviation, 95% CI X.XX, X.XX), cardiovascular (x.xx [x.xx, x.xx]) and respiratory diseases (x.xx [x.xx, x.xx])".

Author summary: "We found that high body mass as well as tobacco smoking in youth were linked with increased risk for diseases of all the studied diseases later in life" - it would make more sense if you deleted "diseases of".

Author summary: Missing full stop on the second bullet point of "What do these findings mean?"

Introduction: The abbreviation OA is still used in paragraph 3 when it has not been properly expanded at first use (in paragraph 2). If you want to use this abbreviation, the text in paragraph 2 should read "…back pain, osteoarthritis (OA), shoulder lesions…". However, the word is written out in full much more often than it is abbreviated to OA, so it would probably be easier to just write it out in full each time.

Methods: Should be "…persons alive and resident in Sweden…", not "…persons alive and residents of Sweden…".

Methods: "…chronic pain conditions that have not well understood etiology…" should be "…chronic pain conditions that have poorly understood etiology…".

Methods: I think the naming of supplementary materials is consistently "S Table #" elsewhere, but here you refer to "S1 Table"

Results: Immediately before Table 3, you refer to the cumulative incidence in 1998-2010. I don't see why you go on to say "until year 2010" - isn't this redundant if you have said it's 1998-2010? There's also a bigger problem. The Methods and Figure 1 both suggest that your outcome is the binary diagnosis of the condition in any of the patient groups (inpatients, outpatients, surgery) between 1987 and 2010 (with only inpatient records available for the whole of this period). Why do you then report cumulative incidence in 1998-2010? The cumulative incidence of a disease will depend hugely on the time period over which it is accumulated, so it's important to make this clear.

Results: Thanks for adding the CI to the paragraph immediately before Table 4. There are some missing spaces in these and a typo in "95%CI 1" to 12%". There are also inconsistent spaces in a few places in "1SD" (I think it should probably have a space).

Table 3: You said that you have added a decimal place of precision to some of the means in Table 3 as I requested, but this has not been done. You have also not added the units of pulse rate.

Figure S1: This is great for showing the (minimal) effect of different adjustment sets. One small point which I missed first time round is that ratios are conventionally plotted on a log scale. That is; you plot log(RR) on a linear scale, but label tick marks with RR (so if your tick marks are at regular intervals, they get closer together at higher values). This applies to your main Figure 2 as well as Figure S1. You've also missed a "d" in "fully ajusted" in the legend. The fact that the choice of adjustment set makes little difference to the estimates is of value to the reader, so in the main manuscript I suggest writing something like "Differences in adjustment variables made little difference to the estimates (S Figure 1)" instead of just "The magnitude of change in the estimates due to adjustments can be seen in S Figure 1".

Figure 1: The second sentence isn't grammatically correct; it should read something like "All exposures were included in the model and they were also adjusted for parental education and occupation". I think some explanation is needed of the points either side of the mean; you could perhaps expand the sentence to read "The middle value on the x-axis corresponds to the mean value (or mode for cardiorespiratory fitness) in the study sample and exposures are plotted over a range of two SD either side of this". The cumulative incidences at the means of these exposures are the same as the 20%, 10% and 3% reported in the third paragraph of the results, but here you refer to them as covering 1987-2010 (which is also what the methods suggest) whereas the earlier report says they are 1998-2010. Please check this. Finally, I think the colours are better than they were but I suspect some colour-blind people would still struggle with them, as well as anyone who printed the paper greyscale. Why not use different styles of point or line to avoid this problem, as you have in Figure 2?

Discussion: It should be "Increased pain sensitivity as a consequence of headache...", not "Increased pain sensitivity as consequence of headache..."

Discussion: There's an unfinished sentence at the end of the penultimate paragraph: "Headache, especially frequent,". Did you mean to continue this or delete it?

Data statement: Should be "national Swedish registers", not "national Swedish registered". Also "included in the supplementary file", not "included in supplementary file".

Conflicts of interest: Should be "AK acts..." not "AK act...".

S Table 3: BMI has been included here, but there's no results for it. It would be useful if the adjusted and unadjusted BMI models could be compared.

S Table 5: BMI should be analysed in SD, like the other exposures (not per 5 units). This shouldn't change the (minimal) effect of adjusting for BMI instead of height and mass, on the other exposures, but it makes it much easier to compare the estimates for BMI itself with those for mass. I found myself making comparisons on the basis of guesswork - I might expect BMI in this sample to have a SD of about 3 units, which would make the RR per SD about 1.13 - very close to your result for body mass. But if you analyse BMI in SD, readers will be able to see this similarity much more clearly.

Supplementary code: This looks a bit incomplete - could you include the code used for the other adjustments sets, the sensitivity analyses, etc? Also, you label a variable called "outocme" (not "outcome"); this would cause an error message if no such variable exists.

---

## [Decision Letter · Decision Letter 3]

21 Nov 2024

Dear Dr Turkiewicz,

Many thanks for re-submitting your manuscript "Physical health in young males and risk of chronic musculoskeletal, cardiovascular and respiratory diseases in middle age – population-based cohort study" (PMEDICINE-D-24-02132R3) for review by PLOS Medicine. The paper has been see again by two of the original reviewers; their comments are included below and can also be accessed here: [LINK]

Thank you for your detailed response to the editors' and reviewers' comments. We invite you to revise the manuscript as discussed via email and according to the points outlined below under editorial and reviewer comments. We plan to send the revised paper to some or all of the original reviewers.

We ask that you submit your revision by Dec 12 2024. However, if this deadline is not feasible, please contact me by email, and we can discuss a suitable alternative.

Don't hesitate to contact me directly with any questions (atosun@plos.org). 

Best regards, 

Alexandra 

Alexandra Tosun, PhD 

Associate Editor

PLOS Medicine

atosun@plos.org

Comments from the editorial team:

We are happy for you to keep the primary analysis as it is. We ask that you (1) provide additional text in the paper explaining your decision not to index, and (2) provide this as a sensitivity analysis, as outlined/requested by the reviewer.

Comments from the reviewers: 

Reviewer #2: Thank you for responding to my comment. The key issue here is power outputs in Watts is not the correct measure of CRF. Calling this variable CRF is not correct. The correct variable for CRF here is W/kg. CRF is generally used to provide an assessment of physical performance. For example in the classic Aerobics Centre Longitudinal Study papers examining the association between CRF and health outcomes, the CRF variable used was treadmill running time as measured in an incremental test. This variable would correspond to W/kg. VO2max in ml/kg/min, but not in l/min, is a strong predictor of running performance. Thus this issue here is that the exposure variable should be measured in W/kg, not W. Virtually all of the epidemiological literature on the association between CRF and health outcomes will express CRF in this way (i.e. running performance, VO2max in ml/kg/min, VO2 max in METS (where 1 MET is 3.5 ml/kg/min), or in W/kg). Including body mass separately in the model and then saying that the W value represents CRF is not correct. If you follow the authors logic to its conclusion, then almost 50 years of epidemiological research on CRF and health outcomes (and even longer examining CRF and exercise performance) is wrong, as is the widely used concept of using METS to express fitness and exercise intensity. It is bold to assert that your approach is correct, and the rest of the field is wrong, and I don't think that the authors have made a convincing argument to that effect. Model A is the correct model to use here, and, as I suspected, when you express CRF in terms of W/kg, a higher level of CRF is associated with lower (rather than higher) risk of MSK disease (as it does for CVD and respiratory disease), and this is true in models with and without resting heart rate. This is the expected finding here and has biological plausibility. The original finding that CRF is positively associated with risk of MSK diseases is an artifact of expressing CRF using inappropriate units. The findings are still worthy of publication with the new (correct) analysis, as presented in model A, with appropriate reinterpration of the implications. Please could the authors reanalyse the data along these lines and also revise the discussion as appropriate.

I understand that this involves a substantial amount of work and that the authors are reluctant to make changes that alter the paper's key message. However, I do feel that this is important to make this correction, and that this paper (with the appropriate data analysis and interpretation) can make an important contribution to the literature. 

Reviewer #3: Thank you for the comprehensive response. You have addressed all the points in my most recent review. There was one thing which I missed previously (sorry) but which I think should be corrected. In most of the supplementary tables and in Table 4, the variable "Stomach problems" has four levels; often, sometimes, occasionally and never (reference category). In S Tables 3-5, the reference category is omitted - this doesn't leave out any important information but I think it should be added for clarity and for consistency with the other categorical variables. Perhaps more importantly in S Table 3 the "occasionally" category is also missing - I'm guessing this is either an accidental omission which should be corrected, or for some reason categories were merged/left out of this analysis (in which case this should be explained in the footnotes). I haven't checked through all the variables in all the tables, but please can you make sure that they are complete?

There is also a missing full stop in the revised notes on Figure 1.

It's for these minor reasons that I've responded with "Minor Revisions" rather than "Accept", but they should be easily sorted out between authors and editor and I don't think I need to see another draft. I look forward to seeing the published article.

---

## [Decision Letter · Decision Letter 4]

13 Dec 2024

Dear Dr. Turkiewicz,

Thank you very much for re-submitting your manuscript "Physical health in young males and risk of chronic musculoskeletal, cardiovascular and respiratory diseases in middle age – population-based cohort study" (PMEDICINE-D-24-02132R4) for review by PLOS Medicine.

Thank you for your continued detailed response to the editors' and reviewers' comments. I have discussed the paper with my colleagues and the academic editor, and it has also been seen again by one of the original reviewers. The changes made to the paper were mostly satisfactory to the reviewer. As such, we intend to accept the paper for publication, pending your attention to the reviewers' and editors' comments below in a further revision. When submitting your revised paper, please once again include a detailed point-by-point response to the editorial comments.

[LINK]

In revising the manuscript for further consideration here, please ensure you address the specific points made by each reviewer and the editors. In your rebuttal letter you should indicate your response to the reviewers' and editors' comments and the changes you have made in the manuscript. Please submit a clean version of the paper as the main article file. A version with changes marked must also be uploaded as a marked up manuscript file. Please also check the guidelines for revised papers at http://journals.plos.org/plosmedicine/s/revising-your-manuscript for any that apply to your paper.

We ask that you submit your revision within 1 week (20 Dec 2024). However, if this deadline is not feasible (including due to the upcoming holidays), please contact me by email and we can discuss a suitable alternative. Please note that the editorial team will be out of office from 23 December 24 up to and including 3 January 25.

Please do not hesitate to contact me directly with any questions (atosun@plos.org). If you reply directly to this message, please be sure to 'Reply All' so your message comes directly to my inbox.  

We look forward to receiving the revised manuscript.

Sincerely,

Alexandra Tosun, PhD

Associate Editor 

PLOS Medicine

plosmedicine.org

Comments from Reviewers:

Reviewer #2: Thank you for including the sensitivity analysis. I understand the argument that maximum power in Watts and body mass have associations with MSD in opposite directions. However maximum power is Watts is not the appropriate measure of cardiorespiratory fitness, it is in fact a measure of cardiorespiratory fitness x body mass. This is because cardioresiratory fitness is related to your ability to move your own body (and therefore your own body mass), not your ability to move an external load. Hence normalisation to body mass is needed. I would be content with the paper if the variable that is currently referred to as "cardiorespiratory fitness" is renamed "maximal power output" (or similar wording). This is more technically correct and gets round the authors' concerns about using a ratio variable. Would the authors be able to make this minor change in terminology. I would be happy to accept the paper if they could do this.

[LINK]

Requests from Editors:

GENERAL

1) Please note that we require authors to add line and page numbers for review purposes. We have added line numbers to the clean version document (Turkiewicz_Physical health_young men_main_rev20241128_clean) and attached the document for your convenience.

2) Please note that references to tables and/or figures (including supplementary material) should be placed before punctuation. This also applies to citations.

3) There appear to be some instances, e.g. l.289, where you use 'cardiovascular fitness' instead of 'cardiorespiratory fitness'. Please check (throughout) and revise if necessary.

4) We have discussed Reviewer #4's final request with the Academic Editor to change the wording from "cardiorespiratory fitness" to "maximal power output" (or similar). The Editorial Team and the Academic Editor suggest adding an additional line in the Methods that mentions maximal power output, but we will not ask you to replace the term "cardiorespiratory fitness" throughout the manuscript.

TITLE

Please revise your title according to PLOS Medicine's style. Your title must be nondeclarative and not a question. It should begin with main concept if possible. "Effect of" should be used only if causality can be inferred, i.e., for an RCT. Please place the study design ("A randomized controlled trial," "A retrospective study," "A modelling study," etc.) in the subtitle (ie, after a colon).

ABSTRACT

1) l.3: Please replace ‘elderly’ with ‘older’. Please revise throughout.

2) l.12: Please define ‘GEE’ at first use.

3) l.14ff: For all numerical results, please re-iterate the statistical information for each pair of parentheses. Fo example: “We found that higher body mass was associated with higher risk of musculoskeletal (risk ratio [RR] per 1 standard deviation [SD] 1.12 [95% confidence interval 1.09, 1.16]), cardiovascular (RR 1.22 [95% CI (1.17,1.27)] per 1 SD)…” Please revise throughout for the Abstract and the main text for consistency.

4) l.16ff: Please note that you have used interpuncts for the next two brackets. Please revise.

5) ll.22-24, please revise for clarity, for example: “Self-reported headache (category often compared to never) was associated with musculoskeletal diseases (RR 1.38 [95% CI (1.21,1.58) per 1 SD] and cardiovascular diseases (RR 1.29 [95% CI (1.07,1.56) per 1 SD], but not with respiratory diseases (RR 1.13 [95% CI (0.79,1.60) per 1 SD].”

6) In the last sentence of the Abstract Methods and Findings section, please describe the main limitation(s) of the study's methodology.

7) Abstract Conclusions: Please use the past tense when describing results, i.e. “While high body mass was a risk factor for all three studied groups of diseases, high cardiorespiratory fitness and high muscle strength in youth were associated with increased risk of musculoskeletal disease in middle age.”

8) Please ensure that all numbers presented in the abstract are present and identical to numbers presented in the main manuscript text.

9) Please include the important dependent variables that are adjusted for in the analyses.

10) We suggest detailing the specific diseases included in the three main groups in the abstract, as there aren't too many. For example, on lines 10-11: We followed the participants through the Swedish National Patient Register for incidence of common musculoskeletal (Shoulder lesions, myalgia, osteoarthritis, joint pain, back pain), cardiovascular (Ischaemic heart disease, atrial fibrillation), and respiratory diseases (Asthma, chronic obstructive pulmonary disease, bronchitis). 

AUTHOR SUMMARY

Please change the final bullet point of 'What Do These Findings Mean?' to: The main limitation is that based on the data source, we were only able to include males in the study.

INTRODUCTION

1) If there has been a systematic review of the evidence related to your study (or you have conducted one), please refer to and reference that review and indicate whether it supports the need for your study.

2) l.71: Please refer to low or middle income countries rather than "developing countries" or "the Global South". 

3) ll.72-74: Please provide references.

METHODS AND RESULTS

1) Table 4: Do the values that show 1.00 represent the reference for the exposure in question? If so, we suggest you clarify this.

2) Figure 1: Please list all exposures in the figure description and define musculoskeletal, cardiovascular and respiratory. Please note that you talk about “exposures” in table 3 and in table 4, so it is important to clarify which set of exposures you refer to. Also, please define ‘SD’ and ‘CI’.

3) Figure 2: Please define ‘CI’.

4) Please briefly mention the results form the sensitivity analyses in the main text (S Table 9 and 10). In general, please make sure that findings from any sensitivity analyses are briefly explained, e.g. whether they support the findings of the main analyses.

5) l.247ff: We have noticed that you do not present findings on all exposures of interest, and that you report findings selectively. For example, findings on height, blood pressure, resting pulse, haematocrit, drug use and alcohol intake are not mentioned at all. We feel that for a balanced presentation, even if no associations were found between the exposures and the three disease groups, this should be clearly stated in the results section. Please revise (throughout).

6) ll.272-274, “Surprisingly, headache seems to present an association with a “dose-response” pattern with both musculoskeletal and cardiovascular diseases, even though the confidence intervals are rather wide.” – please re-phrase and avoid describing the associations as a “dose-response” pattern.

7) ll.248-250, “The associations between the exposures and the different outcomes (disease groups) differ both in magnitude and direction, both in unadjusted (S Table 3) and adjusted analyses (Table 4, S Tables 4, 5, 6).” – We suggest that this should be explained in more detail. The description “differ both in magnitude and direction” is rather vague.

8) ll.287-289: “Most exposures show the same direction and similar magnitude of the associations with all included musculoskeletal diseases.(Figure 2) However, the weakest (or inconclusive) associations are with joint pain and myalgia” – Similar to earlier, these statements seem rather vague and do not really say much about the actual results. It could even be argued that they do not reflect the results shown in the figure. Please revise for accuracy.

9) ll.289-289, “However, the weakest (or inconclusive) associations are with joint pain and myalgia.” – please elaborate. 

10) ll.289-291, “Higher cardiovascular fitness and lower pulse at rest, after adjusting for other exposures, seem to be associated with higher risk of osteoarthritis, while not as clearly with other included musculoskeletal diseases.” – The risk ratio for pulse at rest in the OA group is 0.9 [0.87,0.94], which seems to indicate a lower risk of OA. Please clarify.

11) l.293: “for other specific diseases.” – please specify.

12) ll.293-295, “Alcohol use and smoking, as well as self-reported health measures have generally similar associations with all included musculoskeletal diseases.” – What are these associations? Again, please provide more details and do not report on exposures selectively.

DISCUSSION

General guidance: Please present and organize the Discussion as follows: a short, clear summary of the article's findings; what the study adds to existing research and where and why the results may differ from previous research; strengths and limitations of the study; implications and next steps for research, clinical practice, and/or public policy; one-paragraph conclusion.

1) l.309: Please temper claims of primacy of results by stating, "to our knowledge" or something similar. Please revise throughout (e.g line 394).

2) Please remove any subheadings. The conclusion should be a continuous part of the discussion.

REFERENCES

Where website addresses are cited, please use the word ‘accessed’ when specifying the date of access (e.g. [accessed: 12/06/2024]).

SUPPLEMENTARY MATERIAL

1) Please note that all supplementary files should be references in the main text (e.g. S Table 2 does not seem to be referenced). Please check and revise.

2) In the published article, supporting information files are accessed only through a hyperlink attached to the captions. For this reason, you must list captions at the end of your manuscript file. You may include a caption within the supporting information file itself, as long as that caption is also provided in the manuscript file. Do not submit a separate caption file.

When SI files are contained with a single file:

Please label the file as ‘S1 Supporting Information’.

Please apply alphabetical labelling to each table and figure contained within the S1 file. For example, ‘Fig A’ to ‘Fig Z’ and ‘Table A’ to ‘Table Z’.

Plain text does not need to be labelled and can just be given a title as necessary. For example, ‘Statistical Analysis Plan’.

Please cite tables/figures as ‘Fig A in S1 Supporting Information’ and/or ‘Table A in S1 Supporting Information’, for example.

Please cite plain text as, ‘Statistical Analysis Plan in S1 Supporting Information’, for example.

When SI files are uploaded as separate files:

Please label tables as ‘S1 Table’ (so on) and figures as ‘S1 Fig’ (and so on).

Any additional documents (protocols/analysis plans etc.) can be labelled as ‘S1 Protocol’, for example. Please cite items as exactly as labelled.

General Editorial Requests

---

## [Editor Report · Decision Letter 5]

20 Dec 2024

Dear Dr Turkiewicz, 

On behalf of my colleagues and the Academic Editor, David Flood, I am pleased to inform you that we have agreed to publish your manuscript "Physical health in young males and risk of chronic musculoskeletal, cardiovascular and respiratory diseases by middle age: A population-based cohort study" (PMEDICINE-D-24-02132R5) in PLOS Medicine.

I appreciate your thorough responses to the reviewers' and editors' comments throughout the editorial process. We look forward to publishing your manuscript, and editorially there are only a few remaining minor stylistic points that should be addressed prior to publication. We will carefully check whether the changes have been made. If you have any questions or concerns regarding these final requests, please feel free to contact me at atosun@plos.org.

Please see below the minor points that we request you respond to:

1) Abstract, l.25: For clarity, we suggest writing "Self-reported headache (category 'often' compared to 'never')..." (please note the use of single quotation marks).

2) Abstract, l.27: We are not convinced that 'inconclusive association' is an appropriate description given the CI values presented and ask you to change the sentence to 'no association'.

Before your manuscript can be formally accepted you will need to complete some formatting changes, which you will receive in a follow up email (including the editorial points above). Please be aware that it may take several days for you to receive this email; during this time no action is required by you. Once you have received these formatting requests, please note that your manuscript will not be scheduled for publication until you have made the required changes.

PRESS

Sincerely, 

Alexandra Tosun, PhD 

Associate Editor 

PLOS Medicine